# Human Endogenous Retrovirus K Rec Forms a Regulatory Loop with MITF that Opposes the Progression of Melanoma to an Invasive Stage

**DOI:** 10.3390/v12111303

**Published:** 2020-11-13

**Authors:** Manvendra Singh, Huiqiang Cai, Mario Bunse, Cédric Feschotte, Zsuzsanna Izsvák

**Affiliations:** 1Max-Delbrück-Center for Molecular Medicine (MDC), Helmholtz Society, Robert-Rössle-Strasse 10, 13125 Berlin, Germany; ms3559@cornell.edu (M.S.); huiqiang.cai@clin.au.dk (H.C.); mario.bunse@mdc-berlin.de (M.B.); 2Department of Molecular Biology & Genetics, Cornell University, 526 Campus Road, Ithaca, NY 14853, USA; cf458@cornell.edu

**Keywords:** HERV-K, Rec, LTR5_Hs, MITF, melanoma, proliferative, invasive

## Abstract

The HML2 subfamily of HERV-K (henceforth HERV-K) represents the most recently endogenized retrovirus in the human genome. While the products of certain HERV-K genomic copies are expressed in normal tissues, they are upregulated in several pathological conditions, including various tumors. It remains unclear whether HERV-K(HML2)-encoded products overexpressed in cancer contribute to disease progression or are merely by-products of tumorigenesis. Here, we focus on the regulatory activities of the Long Terminal Repeats (LTR5_Hs) of HERV-K and the potential role of the HERV-K-encoded Rec in melanoma. Our regulatory genomics analysis of LTR5_Hs loci indicates that Melanocyte Inducing Transcription Factor (MITF) (also known as binds to a canonical E-box motif (CA(C/T)GTG) within these elements in *proliferative* type of melanoma, and that depletion of MITF results in reduced HERV-K expression. In turn, experimentally depleting Rec in a *proliferative* melanoma cell line leads to lower mRNA levels of MITF and its predicted target genes. Furthermore, Rec knockdown leads to an upregulation of epithelial-to-mesenchymal associated genes and an enhanced invasion phenotype of *proliferative* melanoma cells. Together these results suggest the existence of a regulatory loop between MITF and Rec that may modulate the transition from *proliferative* to *invasive* stages of melanoma. Because HERV-K(HML2) elements are restricted to hominoid primates, these findings might explain certain species-specific features of melanoma progression and point to some limitations of animal models in melanoma studies.

## 1. Introduction

Endogenous retroviruses (ERVs) are remnants of retroviruses that once infected the germline and became vertically inherited as part of the host genome. Sequences derived from various ERVs account for 8% of the human genome [1], reflecting multiple waves of retroviral invasion in the human lineage. In the human genome, HERV-K(HML2) is the most recently endogenized HERV. Although some HERV-K(HML-2) insertions are polymorphic in the human population [2,3,4,5], suggesting relatively recent integration events, no fully-active provirus has been identified. Due to recombination events and other post-insertional mutations, the vast majority of HERV-K (HML2) members (~950–1200) are either solitary long terminal repeats (solo LTRs) or variously truncated copies [6,7]. A subset of proviral copies (~90) is still capable of expressing retroviral proteins (e.g., Gag, Pro-Protease, Pol-Polymerase, Env-Envelope), and can apparently produce viral particles [8,9,10]. HERV-K(HML2) proviruses have been further classified as type 1 (~26%) or type 2 (~74%), depending on the presence or absence of a 292-bp deletion overlapping the *Pol-Env* junction [11]. Np9 and Rec are alternatively spliced products of Env [8,12]. Type 1 proviruses may only encode Np9 [12], whereas type 2 proviruses may express the RNA isoforms encoding Env, Rec, and Np9 [13,14,15]. Based on the phylogeny of their LTR sequences, HERV-K(HML-2) can be further classified into Long Terminal Repeats (LTR5_Hs), LTR5_A/LTR5_B subfamilies [16,17]. Elements bearing LTR5_Hs represent the youngest integration events in the human genome; with 615 copies (45 proviruses and 570 solo LTRs), 90% of them being exclusive to humans, and absent in other great apes [16]. Type 1 and 2 proviruses are about evenly split among these elements, with 20 and 17 copies, respectively [9].

Rec is a small RNA-binding protein considered to be a functional homolog of the HIV Rev accessory protein [13]. Rec contains a nuclear localization signal (NLS), which in addition to facilitating nuclear transport of the Rec protein, recognizes the Rec-responsive element (RcRE) within the viral RNAs [14,18]. Binding of Rec to RcRE stabilizes incompletely spliced/unspliced viral transcripts and enhance their nuclear export [8,19,20] via the CRM1 export pathway [12,13,21]. Moreover, Rec interacts with Staufen-1 to further facilitate unspliced viral RNA transport [22]. Furthermore, Rec can directly modulate cell signaling via binding to the promyelocytic leukemia zinc finger (PLZF) protein, a transcriptional repressor of the c-MYC proto-oncogene [23]. Rec also binds the testicular zinc-finger protein (TZFP) and the human small glutamine-rich tetratricopeptide repeat protein (hSGT), which are transcriptional repressors of the androgen receptor [22,24]. While it is evident that Rec must have first evolved to facilitate HERV-K replication, there is speculation that the protein may have been co-opted for physiological function(s), such as spermatogenesis [21,25] and defense against viral infections in the early human embryo [26]. Although Np9 is expressed in various types of cancer [12,27] and healthy tissues [28], unlike Rec, it has no identified function promoting the replication of HERV-K [29].

It has been reported that at least 18 HERV-K genomic loci can be transcribed in healthy human tissues and express potentially-coding Rec or Np9 mRNAs [28]. However, the landscape of transcribed HERV-K loci differs considerably between cell/tissue types [28]. Rec RNA/protein expression has been observed in human embryonic tissues, placenta, and retina [15,30]. Early embryonic cells appear favorable to HERV-K expression as proviral mRNAs and proteins are relatively abundant in both germinal- and pluripotent stem cells [26,31]. While HERV-K expression has been observed at low levels in various somatic healthy tissues, it can be further induced by environmental stress, such as ultraviolet irradiation (e.g., UVB and UVC), starvation, or viral infection [32,33,34,35,36]. Furthermore, many studies have reported robust mRNA upregulation of several genomic loci of HERV-K in various disease states, including certain autoimmune diseases and several cancers, especially lung, breast cancers, germ cell tumors, and melanoma [33,37,38] and reviewed in [39,40]. Moreover, transcriptional activation and translation of various HERV-K products [37,41] and even virus-like particles have been observed in specific cancer cells such as teratoma, embryonic carcinoma, and melanomas [8,10,42,43].

Whether and how the overexpression of HERV-K products may contribute to the etiology or progression of these diseases has been the subject of many studies, debates, and speculations. Several mechanisms have been proposed by which HERV-K might contribute to disease. Around 50% of the HERV-K LTRs (including solo LTRs) retained some of their promoter/enhancer activity [44], and are activated in healthy and cancer tissues. The activated LTRs could act as alternative promoters and de-regulate tumor suppressor genes or proto-oncogenes (reviewed in [7,45]). Furthermore, the Env protein of HERV-K, via its fusogenic property, is capable of inducing cell–cell fusion and could, therefore, contribute to tumor invasiveness [27,33,46]. Env could also have oncogenic properties through direct interference with cell signaling pathways (e.g., RSS/MEK/ERK) [47,48]. Indeed, the overexpression of the HERV-K Env protein has been reported to induce an epithelial to mesenchymal transition (EMT)-like process [47,49], a crucial event in oncogenesis leading to the more malignant phenotype. In addition, Env was suggested to promote tumorigenesis via modulation of the immune response [50,51,52]. Confusingly, HERV-K may have either a positive or negative effect on the immune system [53,54]. Similar to Env, the expression of Rec and Np9 is significantly increased in several pathological conditions [55], including germ cell tumors and melanoma [12,21,37,42,43,56,57,58,59,60], but again it remains unclear whether and how these factors could contribute to tumorigenesis and other disease states.

To begin answering this question, we investigate the regulatory genomic aspects of LTR5_Hs sequences, as well as the oncogenic properties of HERV-K Rec (Rec thereafter) in melanoma. We used in vitro models as well as high throughput data mining, including single-cell transcriptome analyses of patient samples, to evaluate the impact of Rec expression in the spatiotemporal progression of melanoma. We found that Rec marks the *proliferative* state of melanoma, and similarly to Env [47] modulates the EMT-like process of cell transformation. However, in contrast to Env [47], Rec appears to inhibit the transition from proliferative to invasive state and as such may represent a protective factor in melanoma.

## 2. Materials and Methods

### 2.1. Generating the Knockdown (KD)-Rec Cell Lines

All 3 KD targeting the ERVK6(HML2.HOM) locus. KD1 targets 3′UTR, KD2 3′UTR, and KD3 the gene body of Rec (Appendix A). KD1 (as) 5-’ATCCATTCAACTCTGAGTGGA-3′, chr7:4,572,718–4,593,717; KD2 (as) 5′-TAAGGCTGACTGTAGATGTAC-3′chr7:4,580,334–4,601,333, chr7:4,599,327–4,599,347; KD3 (as) 5′-CAACGGTGCTCGATTGCGATG-3′ chr7:4,585,379–4,585,399, chr7:4,593,883–4,593,903. Note that due to the high similarity of the HML2 sequences, other loci are also likely to be targeted. The control (scrambled) KD was designed to target GFP cDNA, GCGAAGTACGAATAGTTATCA from [61]. To generate stable KD clones, the targeting sequences were cloned into pT2-MP71-(KD-Scr/1/2/3)-EGFP construct [62], and co-transfected with SB100X transposase [63] into A375 and SKMel-28 cells. The stable lines, which should also express a green fluorescent protein (GFP) reporter, were selected using two-rounds of FACS sorting for the GFP-reporter signal.

### 2.2. Cell Culture

The melanoma cell line, A375 (ATCC CRL-1619) was cultivated in RPMI1640 (Thermo Fisher Scientific, Waltham, MS, USA) supplemented with 10% fetal bovine serum (FBS; PAA Laboratories, Pasching, Austria), 80 U/mL penicillin and streptomycin (Lonza, Basel, Switzerland) at 37 °C in a humidified atmosphere of 95% air and 5% CO_2_. The melanoma cell line, SKMel-28 (ATCC HTB-72) was maintained in Dulbecco’s modified Eagle’s medium (DMEM, Life Technologies, UK) containing GlutaMAX™-I supplement (Gibco, Thermo Fisher Scientific, Waltham, MS, USA, catalog number: 31966-021), with 10% FBS and penicillin and streptomycin in the same condition as above.

### 2.3. Transfection of Melanoma Cells and Cell Sorting

To stably knockdown (KD) Rec, A375, and SKMel-28, melanoma cell lines were transfected with pT2-MP71-(KD1-Rec)-EGFP, pT2-MP71-(KD2-Rec)-EGFP, and pT2-MP71-(KD3-Rec)-EGFP plasmids in the presence of pT2-CAGGS-SB100X, using the Neon transfection system (Life technologies, Carlsbad, CA, USA). The electroporation setting was 1350 V (pulse voltage), 20 ms (pulse width), 2 pulses. Then, 2 × 10^5^ cells in 11 μL resuspension buffer R were combined with 2 μL of purified plasmid mixture (50 ng transposase and 500 ng transposon). Transfected cells were transferred to 6-well tissue culture plates containing 2 mL culture medium. EGFP+ cells were sorted at weeks 1 and 3 post-transfection by FACSAria II cell sorter (Becton, NJ, USA).

### 2.4. Quantitative Real-Time PCR and Semi-Quantitative PCR

Total RNA was extracted using the Direct-zol RNA MiniPrep kit (Zymo Research, Irvine, CA, USA) according to the protocol of the manufacturer. Reverse transcription was performed from 1 µg total RNA with the High Capacity RNA-to-cDNA kit (Applied Biosystems , Waltham, MS, USA). The isolated RNA would be treated with additional DNaseI (Invitrogen, Carlsbad, CA, USA), to get rid of potential DNA contamination. Quantitative real-time PCR (qRT-PCR) was carried out on ABI7900 with PowerUp SYBR Green Master Mix kit (Applied Biosystems, Waltham, MS, USA), according to the recommendations of the manufacturer. For quantification of mRNA expression levels, real-time PCR was run in triplicates for each cDNA sample, using GAPDH and 18 s RNA as the internal controls for SKMel-28 and A375 cells, respectively. For primers and annealing temperatures, see Table 1. Data were analyzed using the comparative CT (2^−ΔΔCT^) method, which describes relative gene expression. For RNA-seq validation, the qPCR control was transfected with pT2-CAG-GFP instead of the KD constructs.

### 2.5. RNA Sequencing and Data Analysis

The concentration of RNA was measured on NanoDrop Spectrophotometer ND-1000, and the quality of RNA was analyzed using Agilent RNA 6000 Nano Kit on Agilent 2100 Bioanalyzer machine. The RNA sequencing library was prepared from 550 ng of RNA, using Illumina (San Diego, USA) TruSeq Stranded mRNA LT Set A kit (Cat. no. RS-122-2101), according to TruSeq Stranded mRNA Sample Prep LS Protocol. Sequencing was performed on an Illumina HiSeq 2000 platform as 100 bp first strand-specific paired-end reads.

Sample-specific barcoded sequencing reads were de-multiplexed from multiplexed flow cells and by using CASAVA. 1.8.2 BCL files were converted to FASTQ format files. The quality of the raw sequence reads was determined by using the FastQC. Reads with a quality score below 30 were removed. We removed the highly variable two nt from the ends of the remaining sequencing reads and mapped them over the reference genome (Human hg19/GRCh37) and transcriptome model (hg19.refseq.gtf). hg19/GRCh37 and hg19.refseq.gtf that were downloaded from USCS tables (http://hgdownload.cse.ucsc.edu/goldenPath/hg19/bigZips/). For mapping the reads, we used our defined settings (i.e., *–alignIntronMin 20 –alignIntronMax 1,000,000 –chimSegmentMin 15 –chimJunctionOverhangMin 15 –outFilterMultimapNmax 20*) to STAR splice mapper. This approach generated STAR genome/transcriptome indices of the reads and provided their respective RefSeq gtf (genes and rmsk) annotations. Final read alignments having more than two mismatches per 100 bp were discarded. We obtained uniquely mapped read counts over individual gene bodies using featureCounts function from the subread package, at gene and transcript level using RefSeq annotations. Scalable gene expression levels were calculated as transcript per million (TPM) counts over the entire gene (defined as any transcript located between transcription start and end sites). In the datasets generated in different batches, we normalized the batch effect using the ‘RUVSeq’ package from R to obtain differentially expressed genes (DEGs). Firstly, we removed the extremely low expressing genes (genes with less than five mapped reads) to avoid the unnecessary variation in transcriptomes. We then used *DESeq2*, an R package, to determine the differentially expressed genes from replicated data. Data were normalized using the default functions of *DESeq2*. The package “*DESeq2”* provides methods to test for differential expression by use of generalized linear models; the estimates of dispersion and logarithmic fold changes incorporate data-driven prior distributions. Genes with an adjusted *p*-value less than 5% (according to the FDR method from Benjamini-Hochberg), Log_2_ Foldchange |1| were assigned as differentially expressed. This strategy provided both quantification and statistical inference of systematic changes between conditions (with at least three replicates). The following samples were RNA sequenced: KD-Scr_A375, KD1-Rec_A375, KD2-Rec_A375, KD3-Rec_A375, KD-Scr_SKMel-28, KD1-Rec_SKMel-28, KD2-Rec_SKMel-28, and KD3-Rec_SKMel-28 in two batches. Canonical pathways and biological function of the differentially expressed genes (DEGs) were further subjected to KEGG and Gene Set Enrichment Analyses (GSEA) using the webgestalt tool.

To estimate expression levels for repetitive elements, we calculated the CPM (counts normalized per million of total reads mappable on the human genome) for each Class II transposable element (TE) locus given in the repeat-masked annotations, downloaded from the UCSC data portal. We had removed those repetitive element’s locus that overlays with the genic exons/UTRs or in the vicinity of 1 Kb from the Transcription Start Site (TSS) of annotated genes before mapping the sequencing reads over the individual locus. To calculate the family-wise expression of repetitive elements, we considered multi mapping reads only if they were mapping exclusively within a repetitive family. Then counted one alignment per read to calculate the CPM. The expression level of repeat families is calculated as Log_2_ (CPM + 1) prior to comparison. The locus-specific information was only calculated for those HML2-HERV-K loci that encode for Rec transcripts. For this, we used “Salmon” that implements the expectation-maximization algorithm to re-distribute and assign the multi-mapping reads to the provided coordinates, based on evidence from uniquely mapped reads.

### 2.6. Single-Cell RNA-Seq Data Processing

We used the single-cell count matrix from GSE72056. We calculated the activity of genes in every cell at TPM expression levels. We considered samples expressing over 1000 genes with expression levels exceeding the defined threshold (Log_2_ TPM > 1). We used Seurat_3.1.1, a package from R to normalize the datasets at the logarithmic scale using “scale.factor = 10,000”. After normalization, we calculated scaled expression (z-scores for each gene) for downstream dimension reduction. We used the original annotations of datasets to classify the Malignant, Non-malignant, and Heterogenous cell-types. To define cell population clusters, we employed the FindClusters function of “Seurat” using “PCA” as a reduction method. The specific markers for each cluster identified by “Seurat” were determined by the “FindAllMarkers” function, using “roc” as a test for significance. This provided us two lists of gene sets. (1) Malignant vs. Non-Malignant differentially expressed genes and (2) genes differentially expressed between Melanocyte Inducing Transcription Factor (MITF)-High and MITF-Low tumors, which we applied for comparison with Rec-KD RNA-seq DEGs. Feature plots, violin plots, and heatmaps were constructed using default functions, except for the color scale that was set manually. The annotated cells were re-clustered using the methodologies described above and visualized on the UMAP coordinates.

### 2.7. ATAC-Seq and ChIP-Seq Data Analyses

ATAC-seq and ChIP-seq raw datasets in the sra format were downloaded from the listed studies and converted to fastq format using sratools function fastq-dump—split-3. Fastq reads were mapped against the hg19 reference genome with the modified bowtie2 parameters: -very-sensitive-local. The modified settings are the following: -D (number of matches) was set to 30, N (number of mismatches in seed) was set to 1, and -R (number of re-seed) was set to 5. All unmapped reads with MAPQ < 10 and PCR duplicates were removed using Picard and samtools. MACS2 called all the ATAC-seq peaks with the parameters—nomodel -q 0.01 -B. Blacklisted regions were excluded from called peaks (https://www.encodeproject.org/annotations/ENCSR636HFF/). To generate a set of unique peaks, we merged ATAC-seq peaks using the mergeBed function from bedtools, where the distance between peaks was less than 50 base pairs. We then intersected these peak sets with repeat elements from hg19 repeat-masked coordinates using bedtools intersectBed with a 50% overlap. To calculate the enrichment over the given repeat elements, we first extended 5 Kb upstream and 5 Kb downstream coordinates from the left boundary (TSS) of respective elements in a strand-specific manner. These 10 Kb windows were further divided into 100 bps bins and tags (bedGraph output from MACS2) were counted in each bin. Tag counts in each bin were normalized by the total number of tags per million in given samples and presented as CPM per 100 bps. We averaged CPM values for each bin between replicates before plotting the figures. To find the transcription factor (TF) binding motifs, 25 bps sequences were extended from either side of MITF ChIP-seq peak summits. The extended sequences were analyzed using the RSAT tool (http://rsat.sb-roscoff.fr/). The TF binding motifs were calculated from JASPAR libraries of human TF motifs. Note that publicly available datasets from primary melanoma cultures contain short reads that are not suitable to identify locus-specific expression of the proviruses. Thus, we used ChIP-seq datasets to find out which loci were regulated through active histone marks and transcription factors.

### 2.8. Cell Invasion Assay

The cell invasion assays were performed using transwell chambers (8 μm pore size; Corning Costar, New York, USA) according to the vendor’s instructions. Briefly, the insert of the wells was first coated with 50 μL BD matrigel. A375 cells were resuspended with RPMI1640 medium containing 0.5% FCS to reach a concentration of 5 × 10^5^ cells/mL. Then, 100 μL cells from each group (stable KD-REC clones, pT2-CAG-GFP control) were seeded into the upper chamber of the insert, with adding DMEM containing 10% FCS to the lower chamber. After incubation at 37 °C in a humidified atmosphere of 5% CO_2_ for 24 h, non-invaded cells were removed with gentle swabbing. In contrast, the invaded ones were stained, and then images were taken with microscopy. Cell numbers were calculated with ImageJ. The control was transfected with pT2-CAG-GFP instead of the KD constructs.

### 2.9. The Accession of Datasets Used in This Study

Encode TF ChiP-seq datasets:ftp://hgdownload.cse.ucsc.edu/goldenPath/hg19/encodeDCC/wgEncodeSydhTfbsGSE60663: H3K27Ac Primary MelanomaGSE60663: H3K4Me3 Primary MelanomaGSE60663: MITF ChIP-seq Primary MelanomaGSE60664: RNA-seq Primary MelanomaGSE46817: RNA-seq of 7 distinct melanoma linesGSE46805: RNA-seq Melanocyte BRAF overexpressionGSE50681: MITF ChIP-seq Melanoma and Melanocyte post-treatmentGSE82330: ATAC-seq Melanoma A375GSE72056: Patient’s Single-cell RNA-seq dataGSE61966: MITF knock-down RNA-seq in 501 Melanoma and MelanocytesGSE50686: COLO829 Melanoma with PLX4032 treatment and Melanocyte

## 3. Results

### 3.1. LTR5_Hs Loci form Active Chromatin in Cancer and Pluripotent Cell Lines

Cis-regulatory elements driving the transcription of the youngest subfamily of HERV-K are embedded within the human-specific LTR5 (LTR5_Hs) [64,65]. Various transcription factors (TFs) have been reported to bind LTR5_Hs, including the pluripotency factors such as OCT4 [66] or the Melanocyte Inducing (also known as MIcrophthalmia-associated) TF (MITF) [65,67]. To examine more systematically the regulation of HERV-K transcription in different cell types, we mined publicly available data produced by the ENCODE consortium [68]. Briefly, we mapped the peaks from 142 ChIP-seq TF datasets from 7 cell lines (ENCODE) over 615 genomic sequences annotated as LTR5_Hs in the human genome (see Methods). Our integrative analysis of various TF occupancy revealed that multiple TFs bound a group of LTR5_Hs loci in a cell-type-specific fashion. Notably, in H1 embryonic stem cells (H1ESCs) and leukemia K562 cells, LTR5_Hs elements tend to be bound by more TFs than in the other cell lines examined (Figure 1A). In the leukemia cell line K562, LTR5_Hs elements are enriched for binding of multiple TFs that control cell cycle and proliferation (e.g., cMYC, MAX, JUND, PU1, and P300, see Figure 1A), which might reflect the transformed phenotype of these cells. Curiously, NANOG binding appeared particularly prominent when H1 cells were cultured in naïve (“3iL”) conditions [69] (Figure 1B).

### 3.2. MITF-Regulated LTR5_Hs/HERV-K Expression Is a Hallmark of the ‘Proliferative’ Type of Melanoma

It has been previously reported that in melanoma, the melanocyte-specific isoform MITF(M) is expressed and promotes the expression of HERV-K transcripts likely through direct binding to LTR5_Hs [67]. To further substantiate that MITF drives HERV-K expression in melanoma, we examined the expression levels of various transposable elements (TEs) in MITF-depleted melanocytes and melanoma cells using previously published RNA-seq datasets [70]. Knocking down MITF affected the expression of small subsets of TE families that were differentially expressed in both melanocytes and melanoma, but there was only a 5% overlap in differential TE expression between the two cell types (Figure 1C,D). Notably, KD MITF was associated with a significant decrease in HERV-K expression (adjusted *p*-value < 0.01, Benjamini–Hochberg (BH) correction) in melanoma, but not in melanocytes (Figure 1C), suggesting that MITF may regulate HERV-K expression specifically in melanomas.

MITF(M) is a critical transcription factor associated with melanoma progression [71,72], and is a sensitive marker of melanoma invasiveness: it is highly expressed (e.g., MITF-High) in the melanocytes and the *proliferative* state of melanoma, but lowly (e.g., MITF-Low) expressed in the *invasive* state [73]. To investigate how tightly HERV-K is controlled in melanoma by MITF, we mined ChIP-seq and RNA-seq data from melanocytes, various melanoma cell lines, and primary melanoma cultures (Methods). Characterizing primary melanoma cultures according to their MITF levels (Figure 1E) revealed an inverse correlation between LTR5_Hs/HERV-K RNA levels and invasiveness (Figure 2A). To see if additional TE families respond to melanoma proliferative/invasive status, we performed a more systematic analysis including other TE families. This analysis revealed that the expression of certain ERVs, including LTR5_Hs/HERV-K, LTR2C, and LTR13, was correlated with that of genes characterizing the *proliferative* state of melanoma (Appendix A). When compared to the overall expression of other TE families, LTR5_Hs/HERV-K was the most abundantly expressed TE family and showed the strongest positive correlation with MITF levels (rho = 0.43, corrected *p*-value < 0.2 × 10^−9^) in the melanoma primary cultures (N = 10) (Figure 2B). These data suggested that similarly to MITF, LTR5_Hs/HERV-K expression correlated negatively with melanoma invasiveness. Next, we characterized the regulation of LTR5_Hs activation in the *proliferative* type of melanoma cells by layering ChIP-seq occupancy signals of MITF and H3K4Me3, H3K27Ac (Figure 2C and Appendix A). Relative to the *invasive* state, the *proliferative* state showed enrichment of LTR5_Hs for these “active” histone marks, coupled with MITF binding (Figure 2C and Appendix A). We found that MITF binding at LTR5_Hs loci coincided significantly with the presence of the canonical E-box motif (CA(C/T)GTG) (*p*-value = 0.026, chi-square test of Observed vs. Expected ratio), which is recognized by MITF as well as c-MYC and MAX TFs [74] (Figure 2C). Overall, our analysis suggests that the upregulation of HERV-K expression in *proliferative* melanoma is likely driven at least in part by direct binding of MITF over the LTR5_Hs through the E-box motif.

### 3.3. Inhibition of BRAF^V600E^ Mutant Leads to the MITF Binding over LTR5_Hs/HERV-K in Melanoma

We also investigated the transcriptional regulation of HERV-K in the background of the BRAF^V600E^ mutation, resulting in a constitutive activation of the mitogen-activated protein kinase (MAPK) pathway [75]. PLX4032, a commonly used small molecule compound in the clinic for the treatment of melanoma, blocks constitutive MAPK signaling via BRAF^V600E^ [76,77]. PLX4032 has been also reported to modify MITF activity and expression in certain melanomas [78], including the highly invasive BRAF^V600E^ mutant COLO829 melanoma line [79], resulting in a MITF-High/AXL-Low phenotype. In COLO829 cells, the MITF global binding profile has a reciprocal pattern at a subset of MITF-binding sites in melanocytes vs. melanoma, while PLX4032 treatment results in re-occupancy along with a switch between invasion types [79]. To examine how PLX4032 treatment may affect HERV-K regulation, we reanalyzed publicly available MITF ChIP-seq datasets of primary melanocytes and COLO829 with and without PLX4032 treatment (GSE50681). We observed that PLX4032 treatment results in a substantial increase of MITF ChIP-seq peaks over LTR5_Hs elements (123 out of 615 loci were bound under PLX4032 treatment but only 19 were bound in the control cells) (Figure 2D), suggesting that MITF binds LTR5_Hs loci, preferentially in BRAF inhibited melanoma cells. Among all TE families in the human genome, the gain of MITF binding under PLX4032 treatment was unique to the LTR5_Hs family (*p*-value < 0.01, chi-square test of observed vs. expected ratio) (Figure 2E). While MITF is also expressed and binds to a subset of LTR5_Hs elements in healthy pigmented melanocytes (n = 33), there was no elevated MITF ChIP-seq signal over LTR5_Hs elements upon PLX4032 treatment of those cells (Figure 2D). Thus, the *proliferative* state of melanoma is associated with the robust and widespread binding of LTR5_Hs by MITF.

Could MITF-bound LTR5_Hs loci function as enhancers for neighbor genes? To answer, we reanalyzed the set of genes that were (i) reciprocally responding to the transition of melanoma invasion types; (ii) upregulated after the PLX4032 treatment along with MITF; (iii) had an LTR5_Hs in their neighborhood (10 Kb from Transcription Start Site (TSS)) that are bound by MITF. Among them, we identified seven genes (e.g., PRODH, BFAR, PTAR1, NRBP2, NCOA7, SLC35A5, and NDUFAB1) (Figure 2F,G and Appendix A). Notably, the role of PRODH has been implicated in various cancers [80,81]. Furthermore, the upstream LTR5_Hs is a validated enhancer of the PRODH gene in embryonic carcinoma cells [82]. Our analysis suggested that PRODH was regulated by the neighboring LTR5_Hs enhancer during the reversion of the invasive melanoma phenotype (Figure 2F,G).

Next, we seek to gain insight into the transcriptional activity of individual HERV-K loci in different melanoma states. Because available RNA-seq datasets were all using short reads, which precludes locus-specific analysis of recent TEs like HERV-K, we turned to chromatin ChIP-seq data (which is more mappable) as a proxy to predict transcriptionally active HERV-K loci [83]. Namely, we used the published dataset for H3K27ac and MITF ChiP-seq (GSE60663) for low passage melanoma cells and also ATAC-seq and H3K27Ac data for the A375 melanoma cell line (GSE82334). Using this approach, we were able to identify twelve loci (nine HERV-K proviruses and 3 solo LTR5_Hs) bound by MITF and/or with active chromatin marks (H3K27ac, ATAC peak) in MITF-High primary cells (but not in MITF-Low cells) (Appendix A). For example, Figure 3A presents an illustration of a region on chromosome 7 that includes the ERVK6 locus, which is structured as two nearly-identical tandemly arranged proviruses [84], duplicated around a centrally located third LTR5_Hs, encoding for apparently intact ORFs [28]. The 5′-flanking LTR5_Hs of the tandem proviruses showed a higher H3K27Ac signal in both *proliferative* MITF-High and A375 cells relative to MITF-Low cells, and also significant ATAC-seq and H3K27ac signals in A375 (Figure 3A), indicating that these loci are preferentially active in the *proliferative* type of melanoma. Moreover, genome-wide analyses of ATAC-seq and active histone marks (H3K27ac and H3K4Me1) in A375 cells further support the notion that multiple LTR5_Hs elements (>50) may function as promoters or enhancers in this melanoma cell line (Figure 3B,C). Collectively, these regulatory genomics data suggest that MITF binding contributes to the activation of multiple LTR5_Hs loci copies (both solo and proviral) in the *proliferative* type of melanoma.

### 3.4. Depleting HERV-K-Rec May Induce an EMT-Like Process in A375 Melanoma Cells

From the nine identified actively transcribed proviruses, five were reported to express Rec, from which three were also detected in various healthy tissues [28]. To gain insight into a potential role of Rec in melanoma, we first used an RNAi approach to deplete Rec expression in two melanoma cell lines (A375 and SKMel-28). A375 and SKMEL-28 both carry the BRAF^V600E^ mutation, but A375 has a higher proliferation rate, whereas SKMEL-28, similar to COLO829, is more invasive [85], indicating their distinct invasion phenotypes. SKMel-28 also differs from A375 as it does not express MITF at significant levels [78,86]. To knockdown (KD) Rec, we designed three RNAi constructs (Appendix A) to target the sequence of the ERVK6 locus, for which we observed marks of high transcriptional activity in MITF-High and A375 melanoma cells (Figure 3A). Nevertheless, due to the high level of Rec sequence similarity among several HERV-K proviruses [15], we predicted our KD constructs would target Rec transcripts produced by most transcriptionally-competent HERV-K proviruses [15,28]. In addition, we generated control cell lines expressing scrambled sequences. In total, we generated 10 stable cell lines (6 KD and 4 controls) using *Sleeping Beauty* as a stable vector [62,63]. Real-time qRT-PCR experiments showed that Rec transcript levels were depleted in each of the six independent KD cell lines isolated for A375 and SKMel-28 (Figure 3D). We also observed that knocking down Rec also led to an increased level of alternative HERV-K transcripts encoding Gag and Env in SKMel-28 (all three KD lines) and in A375 (in the A375-KD1 line only) (Figure 3E).

To detect transcriptome changes upon Rec KD, we used RNA-seq to compare the stable KD-Rec lines to their respective control lines. While SKMel-28 expresses Rec from three genomic loci [43], we were able to unambiguously identify five Rec-expressing loci in A375 (Figure 3F). Next, we computed the differentially expressed genes (DEGs, |2|-fold and FDR < 0.05) between the Rec KD and their control lines (Methods). We obtained 171 and 773 DEGs upon Rec KD in SKMel-28 and A375 respectively (Figure 4A,B, Appendix A, see Methods). In SKMel-28, the 171 DEGs were not significantly enriched for any Gene Ontology (GO) categories. In contrast, the 773 DEGs in A375 showed significant enrichment for several GO terms including *pigmentation*, *cell migration, IL1 production* and *cell proliferation* (FDR < 0.05) (Figure 4A).

Notably, we identified a set of genes involved in epithelial-mesenchymal transition (EMT)-like process, which plays a critical role in phenotype switching from the *proliferative* to *invasive* state [87,88,89]. Intriguingly, this set of DEGs included matrix metallopeptidase 2 (MMP2), a classical marker of cell invasion along with additional genes previously reported to drive the canonical EMT process [90], such as a zinc finger protein (SNAI2), forkhead box C2 (FOXC2), cadherin 2 (CDH2) and goosecoid Homeobox (GSC) (Figure 4B). Further, the genes upregulated upon Rec KD in A375 cells included factors such as WNT5B and GSK3B, which are known to induce the EMT-like process and are activated when melanoma gets relapsed to metastasis after failed treatment (Figure 4B) [91,92]. During melanoma metastasis, a TF regulatory network characterized by SNAI2^low^/ZEB2^low^/ZEB1^high^ is established [93,94,95]. The same TF expression signature is observed upon KD of Rec in A375 cells (Figure 4B). Beside canonical EMT markers, we also observe a strong upregulation of several members of the SPANX (sperm protein associated with the nucleus in the X chromosome) family of genes upon Rec KD in A375 cells (Figure 4B). The expression of SPANX genes is normally confined primarily to the germline, but upregulation of SPANXB1/2 has been previously reported in invasive melanoma [96]. Furthermore, we detected a robust decrease in the transcript level of MITF(M), the master regulator of the melanocyte differentiation program, *upon Rec* KD in A375 cells (Figure 4B). In addition to MITF, we also observed a decreased expression of other melanocyte differentiation markers (e.g., TYR, PMEL) [97] (Figure 4C). Together, the transcriptome changes we observed upon Rec KD in A375 cells suggests that Rec expression is part of a regulatory cascade inhibiting the EMT-like process.

To corroborate these results, we chose two of the most DEGs in our transcriptomic analysis (SPANXB1/2 and MITF(M)) and performed RT-qPCR to compare the mRNA levels of these two genes in the three independent Rec KD A375 cell lines and their control lines. The results were consistent with the RNA-seq analysis: SPANXB1/2 was strongly upregulated (Figure 4D) and MITF(M) was robustly downregulated upon Rec KD (Figure 4E).

To substantiate the connection between MITF and Rec, we also determined the level of Rec expression in MITF-depleted cells (GSE61966). Importantly, the expression of Rec was downregulated upon knocking down MITF in the melanoma cell lines (Figure 4F). Our analysis revealed that, upon MITF depletion, the overall expression of Rec-coding loci, including ERVK6, was down (Figure 4F), indicating that MITF and Rec expression levels are tightly associated in melanoma cells. Following the lead on potential MITF and Rec co-regulation, we also asked whether the MITF-target genes (ChIP-seq peaks around the TSS) were among the 773 DEGs in our Rec-KD dataset. This approach identified a set of 167 MITF target genes, whose expression was significantly depleted upon KD Rec (*p*-value = 2.654 × 10^−14^, chi-square test) (Figure 4G), suggesting a reciprocal coregulation of MITF and Rec expression.

### 3.5. Depletion of Rec Results an Enhanced Cell Invasion in A375 Melanoma

To further explore the idea that Rec KD in A375 cells recapitulates part of the expression profile of *invasive* melanoma, we examined the correlation between DEG upon Rec KD and those differentially expressed between *invasive* and *proliferative* primary melanoma cultures. We found a significant positive correlation pattern between the two sets of DEGs (Spearman’s rho = 0.20, *p*-value < 0.05, asymptotic t approximation) (Figure 5A). Overall, these analyses suggest that depletion of Rec in A375 cells may result in an increased invasion phenotype.

To test whether Rec expression modulates the invasiveness of melanoma cells, we performed a trans-well invasion assay to compare the invasiveness of Rec KD A375 cells to their control lines. The results showed that all three Rec KD lines exhibited significantly elevated invasiveness (Figure 5B,C). These data support the hypothesis suggested by the transcriptome analyses that Rec may function as a suppressor of the EMT-like transition process, which modulates the invasiveness of melanoma.

### 3.6. Comparison of the Rec KD A375 with the Malignant Cells in Melanoma Patients

Next, we sought to examine whether a set of dysregulated genes in Rec KD A375 cells were also differentially expressed in malignant cell types. To assess this, we first performed a single-cell (sc) RNA-seq analysis (n = 4645 cells) derived from melanoma patients (n = 19), which includes both malignant and non-malignant cells [98]. Consistent with the original study, our analysis distinguishes the *malignant* from the *non-malignant* cells based on the expression of the most variables genes (Figure 6A) and identifies four major *malignant* cell types, marked by MAGEC2, MITF, APOE (Apolipoprotein E), and SPP1 (Osteopontin) expression, respectively (Figure 6A–C). It is known that high levels of MITF, APOE, and SPP1 mark different aspects of melanoma progression, i.e., melanocyte differentiation, angiogenesis [99], and inflammation [100], respectively. Interestingly, those three genes were all significantly downregulated in the Rec KD A375 cells (Figure 4B and Figure 7A), suggesting that the KD of Rec in those cells recapitulates multiple phases of melanoma.

Having characterized the expression profile of *malignant* and *non-malignant* cells from melanoma patients, we next compared the ~700 DEGs (Figure 7B, see Methods) between *non-malignant* and *malignant* cell types to those detected in our Rec KD A375 experiments (n = 773). There were 94 overlapping genes, showing a significant enrichment over a random expectation. Genes showing malignant-specific expression tend to be upregulated in KD-Rec (Figure 7C).

To substantiate that Rec regulated genes are overrepresented in the obtained cell-types within *malignant* or *non-malignant* clusters, we performed an additional analysis, and use only the top 50 marker genes of each cluster (both malignant and non-malignant) that overlaps with differentially expressed genes upon Rec-KD (Figure 7D). This analysis detects the markers of the MITF overexpressing cell clusters at the highest frequency among the down-regulated genes upon Rec KD in A375 (Figure 7E).

Lastly, we sought to assess whether genes predicted as downstream targets of MITF from the same scRNA-seq analysis [98], were also affected in our Rec KD experiments. We found that 41 out of the top 100 MITF predicted downstream targets were included in our list of 773 DEGs, a significant enrichment over the random expectation [*p*-value, test]. Strikingly, 39 out of the 41 overlapping genes were downregulated by Rec KD (*p*-value < 2.2 × 10^−16^, chi-square test of observed vs. expected). These observations are consistent with the finding that MITF is downregulated in our Rec KD lines, thereby affecting MITF downstream targets (n = 41 out of 100, Wilcoxon-test, *p*-value < 1.2 × 10^−11^) (Figure 7F). This analysis suggests that the downstream targets of MITF might be depleted as the consequence of MITF downregulation upon KD Rec. Thus, our KD Rec A375 model recapitulates MITF-associated aspects of phenotype switching in melanoma patients to some degree.

## 4. Discussion

HERV-K resembles “complex” retroviruses such as HIV-1 by its ability to encode multiple proteins, including Rec. While Env has the typical structure of a transmembrane protein (e.g., extracellular, transmembrane, and cytoplasmic domains), Rec lacks the domains required for a membrane protein function (Figure 8A). Still, it carries nuclear localization/export signals [8,19]. Although Env and Rec are different in sequence and function, our results suggest that Rec, like Env [47], exerts a modulatory effect on the EMT-like process in cancer progression. However, the two proteins appear to act in opposite direction. While Env overexpression (both RNA and protein) may promote tumor progression through ERK1/2 activation [47], our KD experiments indicate that Rec may inhibit the EMT process and oppose melanoma metastasis. Interestingly, similar to Env, expression of the Gag protein encoded by HERV-K, also increase the tumorigenic potential of melanoma cells [46]. Because we observed that KD Rec leads to the upregulation of Gag and Env in one of the three KD lines we established using three different RNAi constructs, we cannot exclude that some of the transcriptomic and phenotypic (invasiveness) changes we observed are due to indirect effects on Gag and Env levels. Importantly, however, we found that all three KD lines behaved similarly in terms of some of the key differences in gene expression (e.g., MITF(M), compare KD lines in Figure 4E) or invasiveness (Figure 5). Thus, it appears that Rec’s oncogenic potential may oppose that of Env or Gag proteins encoded by HERV-K, and when HERV-K is overexpressed, the balance between these different products may be key to maintain cellular homeostasis.

In about 60% of melanomas, the protein kinase BRAF is mutated (BRAF^V600E^) [75]. The BRAF^V600E^ mutation happens at an early stage of the oncogenic process, and is already reported from benign naevi [102]. MITF is considered a crucial factor in melanoma invasiveness [103,104] and plays a critical role in controlling BRAF^V600E^ melanoma [105,106]. Accordingly, the expression of MITF has been widely used as a marker to distinguish between the *proliferative* and *invasive* types of melanoma cells [86,103,105,107]. Reduced MITF expression generates *invasive* melanomas, with metastasis-promoting properties [105].

Our data suggest that MITF activates LTR5_Hs in melanoma (not in melanocytes); thus, Rec expression specifically marks the *proliferative* type of melanoma cells. If Rec expression is involved in maintaining a delicate balance between cell proliferation and invasion, as this study suggests, then Rec activation might be used as a sensitive marker to distinguish between the *proliferative* vs. *invasive* phenotypes of melanoma.

Of note, the application of single-cell analysis and advanced genomic technologies has refined our understanding of melanoma progression, and, instead of the *invasive* and *proliferative* states, suggested a melanocyte differentiation gradient [104]. It was also observed that the chromatin landscape differs between the phenotypes, and for MITF to drive the transcription of proliferation and differentiation genes, the transition might require additional (e.g., epigenetic) changes [73,108].

In addition to the Rec-expressing loci, MITF targets several solo LTR5_Hs that could also function as cis-regulatory sequences (enhancers, promoters) for nearby genes [64]. Notably, some of these LTRs harbor the E-box CA(C/T)GTG motif, which is also recognized by cMYC and MAX oncoproteins [72], suggesting a role of HERV-K-derived regulatory sequences in other oncogenic processes as well. Because several LTR5_Hs elements are insertionally polymorphic in the human population [2,3,4,5], this type of variation could also have implications in melanoma progression and deserves future work.

Under healthy physiological conditions, apparently, a limited number (n = 18, or often less) of HERV-K loci is active [28], whereas, in melanoma patients and/or cell lines, the number of transcribed HERV-K loci is higher (n~24) [43]. These studies indicate that there are specific HERV-K loci transcribed under pathological conditions. Although forced expression of MITF(M) in non-malignant cells has been shown to enhance the levels of various HERV-K-derived transcripts (e.g., *Gag,*
*Env,* and *Rec*) [67]*,* the activated HERV-K loci might differ in both the level of expression and composition of their HERV-K-derived transcripts. Furthermore, a variable set of HERV-K loci seems to be expressed in different melanoma patients [43,109], suggesting that variability in the amount of HERV-K products may contribute to melanoma heterogeneity [37,42], and possibly the progression and outcome of the disease.

Our study indicates that the depletion of Rec in BRAF^V600E^ mutant melanoma cells (A375) results in reduced MITF levels. Thus, our data suggest a model whereby a MITF-dependent regulatory loop is disrupted upon Rec KD through (i) decreased MITF binding to LTR5_Hs, which leads to (ii) reduced Rec levels and (iii) upregulation of EMT markers, contributing to (iv) escalated cell invasion (Figure 8B). Future research is required to decipher the mechanism by which MITF is downregulated upon Rec KD. It would also be interesting to find out whether Rec overexpression can lead to enhanced MITF expression in *invasive* melanoma, and revert the *invasive* phenotype to a *proliferative* one. This would answer whether the impact of Rec on melanoma proliferation status is correlative or (directly or indirectly) causal.

While the in vitro melanoma cultures are not able the fully reproduce melanoma progression, the various melanoma lines might mimic different aspects of it. The COLO829 line represents an *invasive* type of melanoma that is still convertible (upon PLX4032 treatment) to a *proliferative* phenotype, whereas the A375 line is a relatively good model of the MITF expression-associated features of a *proliferative* type of melanoma.

It has been widely discussed how precisely melanoma can be recapitulated in animal models (reviewed in [110]). Interestingly, the analysis of single-cell transcriptome data from multiple human patients suggests a distinct separation of *proliferative* and *invasive* types of cells within a patient [5]. However, these two phases do not co-occur in mice [98,111] As Rec (and HERV-K(HML2)) is hominid-specific, our study might explain certain species-specific features of melanoma progression in humans, and stress the potential limitations of using animal models in melanoma studies.

## Figures and Tables

**Figure 1 viruses-12-01303-f001:**
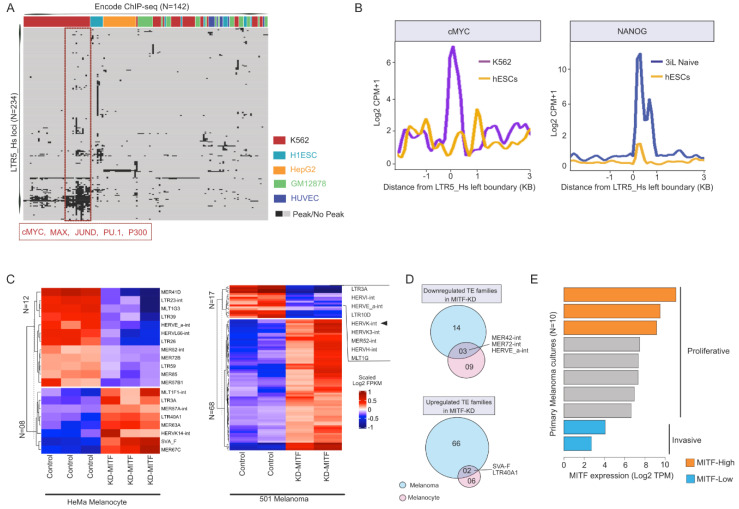
(**A**) Heatmap is summarizing the transcription factor (TF) occupancy (ChIP-seq peaks, ENCODE) over individual Long Terminal Repeats (LTR5_Hs) loci in five human cell lines. This plot includes 234 out of 615 LTR5_Hs loci annotated in the human Refseq genome, which is occupied by at least one ChIP-seq peak. Each row represents an individual locus of LTR5_Hs. Each column shows TF ChIP-seq peak occupancy in a different cell line. Various TFs encompassing LTR5_Hs loci are manually annotated in a given cell line. The cluster of TFs binding over a subset of LTR5_Hs loci is shown in the red box. (**B**) Line plots showing the distribution of normalized ChIP-seq signal counts of cMYC (left panel) and NANOG (right panel) over the annotated LTR5_Hs sequences in the human genome. The comparison is made between primed hESCs (H1) and K562 cells (left panel), as well as primed hESCs (H1) and hESCs (H1), cultured in 3iL “naive” medium (right panel). The ChIP-seq signal counts are calculated relative to primed ESCs, in a 3 Kb genomic window at the left boundaries of LTR5_Hs in the human genome. (**C**) Heatmap displays the expression of differentially expressed TE families (at counts per million level) upon Melanocyte Inducing Transcription Factor (MITF) knockdown (KD) in melanocytes (left panel) and melanoma (right panel). (N) denotes the number of differentially expressed transposable element (TE) families (*DESeq2* adjusted *p*-value < 0.05). Note that HERVK expression is affected in melanoma (marked), but not in melanocytes. (**D**) Venn diagrams illustrating shared and unique differentially expressed TE families between MITF-depleted (KD-MITF) melanocytes and melanoma cells. The upper panel shows down-regulated TEs, whereas the lower panel shows up-regulated TEs upon KD MITF. (**E**) Characterizing primary melanoma cultures according to their MITF expression levels. Bar plots showing the level of MITF expression (Log_2_ TPM) in primary melanoma cultures (n = 10). According to their MITF expression level, cultures are defined as MITF-High (*proliferative*) and MITF-Low (*invasive*) melanoma primary cultures. The annotations indicating the *proliferative* and *invasive* states are taken from the original study [35]. Note that in the following analyses we used the data of MITF-High and MITF-Low as *proliferative* (n = 3) and *invasive* (n = 2). TPM—transcript per million.

**Figure 2 viruses-12-01303-f002:**
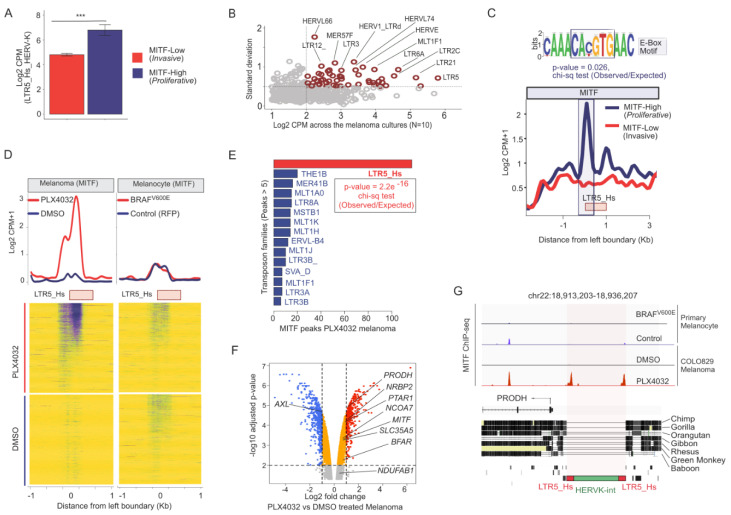
(**A**) Bar plot showing the LTR5_Hs_HERV-K expression (Log_2_ CPM) levels in *proliferative* melanoma (MITF-High) and *invasive* melanoma (MITF-Low) cell lines (GSE60664). *** denote the significance of the differential expression (*p*-value < 0.05, t-test) (**B**) Scatter plot visualizes the average expression (Log_2_ CPM) (*X*-axis) and standard deviation from the mean (*Y*-axis) of transposable element (TE) families in various primary melanoma cultures (N = 10). (**C**) Line plot showing the MITF ChIP-seq signal occupancy in *proliferative* melanoma (MITF-High) and *invasive* melanoma (MITF-Low) cell lines averaged over all of the LTR5_Hs loci (N = 615). Note the E-Box motif, identified in the sequences of LTR5_Hs loci with MITF peaks. The E-Box motif was present in 57 out of 115 MITF bound LTR5_Hs sequence in A375 (*p*-value indicates the probability of the presence of the E-Box motif in MITF bound vs. unbound LTR5_Hs loci determined by Chi-square test). (**D**) The effect of PLX4032 treatment on MITF binding at LTR5_Hs loci. Upper panel: Line plot showing the MITF ChIP-seq signal averaged over LTR5_Hs loci in primary melanocytes and melanoma cells with and without the treatment of PLX4032. Lower panel: Heatmap of MITF ChIP-seq signals. Note: MITF occupies LTR5_Hs genomic loci upon PLX4032 treatment, specifically in melanoma (123 unique loci were bound, but not in melanocytes (over 50 loci). (**E**) Bar plot showing the number of MITF peaks detected over the genomic loci of various TE families in melanoma cells treated with PLX4032. This plot shows only those families which have at least five peaks over the individual locus. (**F**) Volcano plot between Log_2_ foldchange and -Log_10_ adjusted *p*-value shows the differentially expressed genes (DEGs) in COLO829 cells upon the treatment of PLX4032. While AXL is downregulated, MITF and selected set of genes neighbor to LTR5_Hs loci are upregulated, and hence are annotated on the plot. (**G**) Integrative genome browser tracks showing the MITF ChIP-seq signals at the PRODH locus on chr22. The phylogenetic conservation of the locus is demonstrated on the middle panel. The bottom panel shows the schematic structure of the LTR5_Hs/HERVK provirus in *repeatmasker* track. CPM—counts normalized per million of total reads mappable on the human genome.

**Figure 3 viruses-12-01303-f003:**
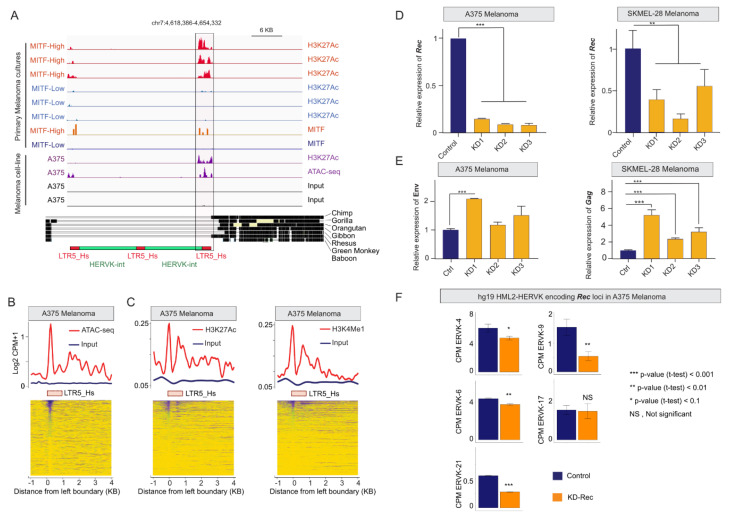
(**A**) Genome browser tracks showing the H3K27Ac, ATAC-seq, and MITF signals at specific LTR5_Hs-HERV-K sequences at the ERVK6 locus on chr7, encoding for Gag, Env, and known to produce Rec (UniProt). Note (i) that ERVK6 is a duplicated, tandemly arranged provirus, flanked by two LTR5_Hs sequences, duplicated around a centrally located third LTR5_Hs (see also Appendix A); (ii) that the signals are more robust in both the MITF-High primary melanoma cultures and A375 cells; the phylogenetic conservation of the locus is demonstrated on the middle panel. The schematic structure of the tandemly repeated locus is shown at the bottom. (**B**) Normalized ATAC-seq (nucleosome-associated) signals averaged over the annotated LTR5_Hs sequences in the human genome in A375 cells. (**C**) Normalized ChIP-seq signals of H3K27Ac (left panel) and H3K4Me1 (right panel) histone marks (corresponding to active chromatin states) averaged over the annotated LTR5_Hs sequences in the human genome in A375 cells. Note that the analysis shows only loci that are mappable from the analyzed ChIP-seq data (likely underestimated). (**D**) Knocking down (KD) Rec in A375 and SKMel-28 cells using RNA interference. Stable KD lines were selected in two-rounds by FACS sorting for the GFP-reporter signal. Bar plots are showing the normalized RT-qPCR quantification of Rec mRNA levels with three KD-Rec constructs. Control, KD-Scr (scrambled). (**E**) Knocking down Rec affects the transcription of alternative HERV-K products, primarily in SKMel-28. Bar plot is showing the effect of knocking down Rec (KD-Rec) on the expression of alternative HERV-K products, determined by real-time qPCR. Note: KD-Rec led to an increased level of Gag in SKMel-28 (all the three biological replicates), whereas Env in A375 (KD1 only). Control, KD-Scr (scrambled). (**F**) Five HERV-K loci express Rec in A375. The bar plots demonstrate the locus specificity of the RNAi strategy against Rec. Relative quantification of Rec expression (CPM) upon KD at specific genomic loci, compared with scrambled control (KD1-3 vs. KD-Scr), showing the significantly affected Rec expressing loci in A375 cells. (* *p*-value < 0.1, ** *p*-value < 0.01, and *** *p*-value < 0.001, t-test). Note that the given mapping strategies of RNA-seq datasets over repetitive sequences the expression level upon KD Rec might not be directly compared with qPCR results (see Figure 3D).

**Figure 4 viruses-12-01303-f004:**
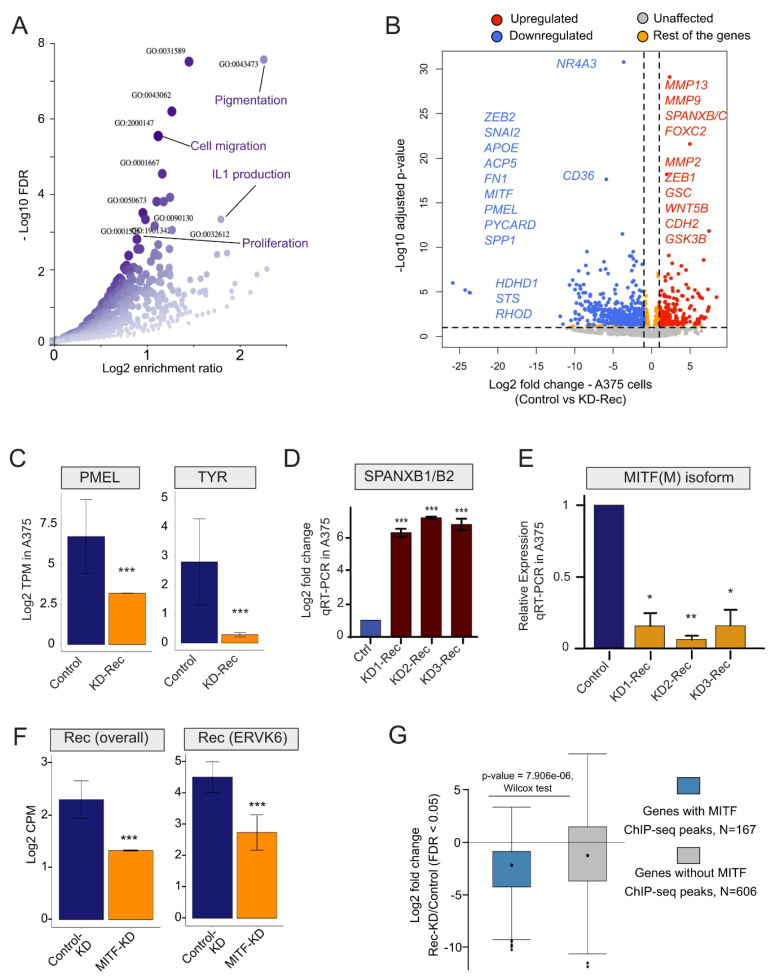
(**A**) Significant Gene Ontology (GO) Biological Process terms enrichment analysis of (differentially expressed genes) DEGs in Rec KD A375 vs. KD-Scr_A375 (scrambled control). (**B**) Volcano plot showing the comparison of Log_2_ fold-changes (*x*-axis) and adjusted *p*-value (Log_10_, *y*-axis) of DEGs upon KD-Rec vs. KD-Scr in A375 cells. The annotated genes are among the top dysregulated genes that are known to mark either the *invasive* (red) or the *proliferative* (blue) melanoma. Red, upregulated; Blue, downregulated; Grey, Unaffected; Orange, Res of the genes; (**C**) Bar plots representing means ± SEM from RNA-seq expression values (Log_2_ TPM) for melanocyte markers (e.g., TYR and PMEL) upon KD-Rec in A375 vs. KD-Scr control. *** denote the significance of differential expression (*p*-value < 0.05, t-test) (**D**) Validating depleted SPANXB1/B2 expression (previously detected as DEG in RNA-seq experiments) upon Rec KD in A375 cells vs. control (Log_2_ fold change), using real-time qPCR (see Methods). (**E**) Validating depleted MITF(M) expression (previously detected as DEG in RNA-seq experiments) upon Rec KD in A375 vs. control (Log_2_ fold change, (* *p*-value < 0.05, ** *p*-value < 0.01, t-test)), using real-time qPCR (see Methods). (**F**) Bar plot representing means ± SEM of transcript expression (Log_2_ CPM) of Rec coding loci (left panel) or the ERVK6 locus upon MITF-KD vs. Control-KD (right panel (GSE61966). *** denote the significance of differential expression (*p*-value < 0.05, t-test). (**G**) Box plot showing the differential expression of genes that were bound or not by MITF in PLX4032 treated melanoma within the set of 773 significant DEGs upon Rec-KD (adjusted *p*-value < 0.05).

**Figure 5 viruses-12-01303-f005:**
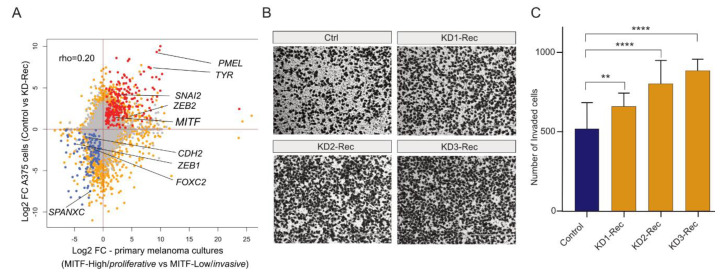
(**A**) Scatter plot showing the pairwise comparison of DEGs between primary melanoma cultures and A375 cells. Primary melanoma culture data derive from the comparison between MITF-high (*proliferative*) and MITF-low (*invasive*) type of melanomas. A375 data derive from the comparison between KD-Rec and KD-Scr (scrambled control). The annotated genes are affected in both comparisons in a similar fashion. Note (i) that MITF-Low (*invasive*) melanoma and KD-Rec share the upregulation of inflammation-associated genes; (ii) The downregulated genes are classical markers of *proliferative* melanomas. Red, Upregulated in both (N = 233); Blue, Down-regulated in both (N = 120); Grey, Unaffected in both; Orange, Rest of the genes. (**B**) Representative inverted light microscope images of trans-well invasion assay performed in A375 melanoma cells upon knocking-down Rec by three KD constructs (KD1-3). The Ctrl expresses pT2-CAG-GFP instead of the KD constructs. Scale bars, 400 μm. (**C**) Quantification of cells invading in the trans-well invasion assay. ImageJ counted the number of invaded cells. Columns indicate the average number of invaded cells per field from two independent experiments. Error bars represent the means ± s.d. of two independent experiments. Mann–Whitney test; ** *p* ≤ 0.01, **** *p* ≤ 0.0001 (See Methods for details).

**Figure 6 viruses-12-01303-f006:**
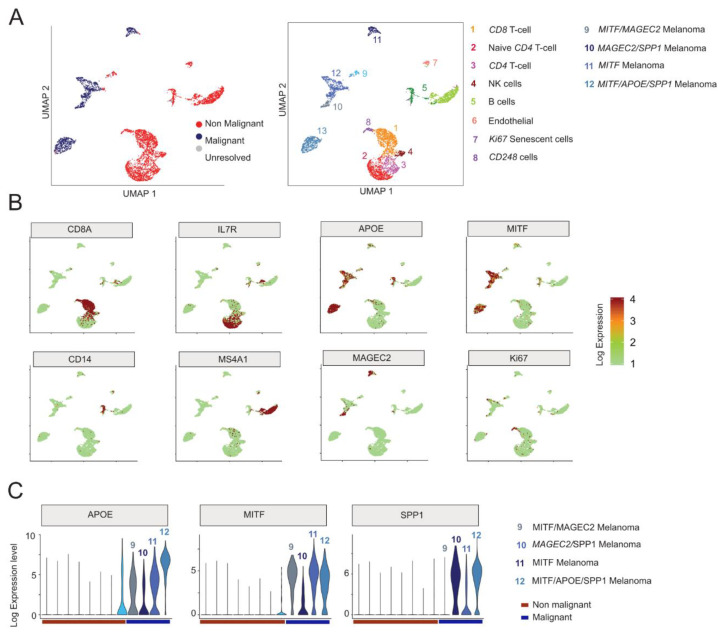
(**A**) UMAP visualization of the twelve distinct cell clusters identified by 10XGenomics scRNA-seq in patient-derived melanoma [98]. The identified clusters are annotated as *Malignant* and *Non-Malignant* (left panel) or according to their origin (right panel), following original annotations [98]). Note that (i) despite their proximity to malignant cell clusters, CD248-labelled cells in Custer9 are most probably *Non-Malignant* cells [101]; (ii) this approach distinguishes four *Malignant* clusters (e.g., Clusters 9–12). (**B**) Feature plots based on the UMAP plot shown on (**A**) visualizing the expression of selected markers used to identify the distinct cell-types of melanoma. The Color intensity gradient indicates the expression of the depicted marker gene. Each dot represents an individual cell. Dots in green denote lower, whereas in dark-red, a higher level of gene expression in a given single cell. (**C**) Violin plots visualize the density and distribution of expression of distinct marker genes in *Malignant* melanoma clusters (9–12).

**Figure 7 viruses-12-01303-f007:**
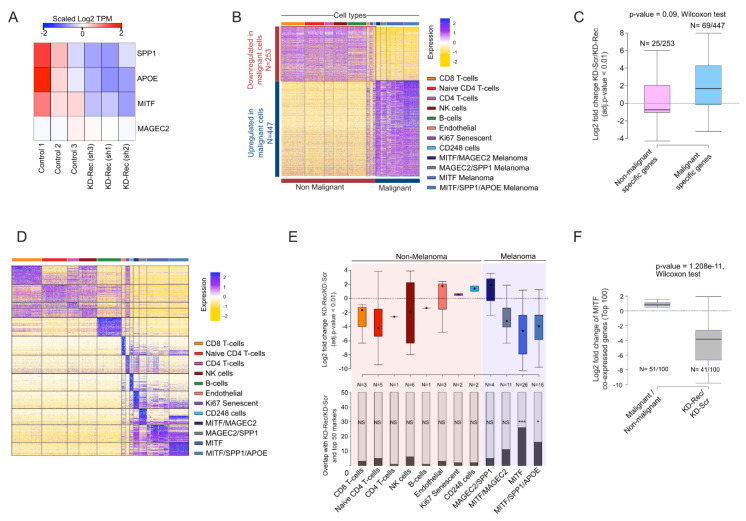
(**A**) Heatmap visualization of scaled (Log_2_ TPM) expression of SPP1, MITF, APOE, and MAGEC2 upon knockdown (KD) of Rec in A375 cells (three replicates). (**B**) Heatmap visualization of scaled expression [log TPM (transcripts per million)] values of a distinctive set of ~1500 genes (Log_2_ fold change >1 and >50% of cells expressing either in *Non-malignant* or *Malignant* cells), which are differentially expressed in scRNA seq data. The color scheme is based on Z-score distribution from −2.0 (gold) to 2.0 (purple). Top color bars highlight representative gene sets specific to the respective clusters. (**C**) Box plot showing the comparison of differentially expressed gene sets in *Malignant* and *Non-malignant* cells (patient-derived scRNA-seq) and KD-Rec vs. KD-Scr in A375. (**D**) Heatmap showing the expression dynamics of the top 50 marker genes for each cluster. Color scheme and annotation as in Figure 7B. (**E**) Lower panel: Stacked barplot showing the number of marker genes that are detected among the DEGS upon Rec KD in A375. Stars indicate the significance (*p*-value < 0.01, two-sided fisher-exact test) Upper panel: Differential expression of overlapping genes are represented at Log_2_ fold change scale upon Rec KD in A375. The color scheme of each box represents the corresponding cell clusters. (as in Figure **7**B). (**F**) Box plot showing the comparison differential expression of a gene set of “MITF-High” (consisting of MITF itself and the top 100 genes correlated with MITF expression) between *Malignant* and *Non-malignant* cells (patient-derived scRNA-seq) and KD-Rec vs. KD-Scr in A375.

**Figure 8 viruses-12-01303-f008:**
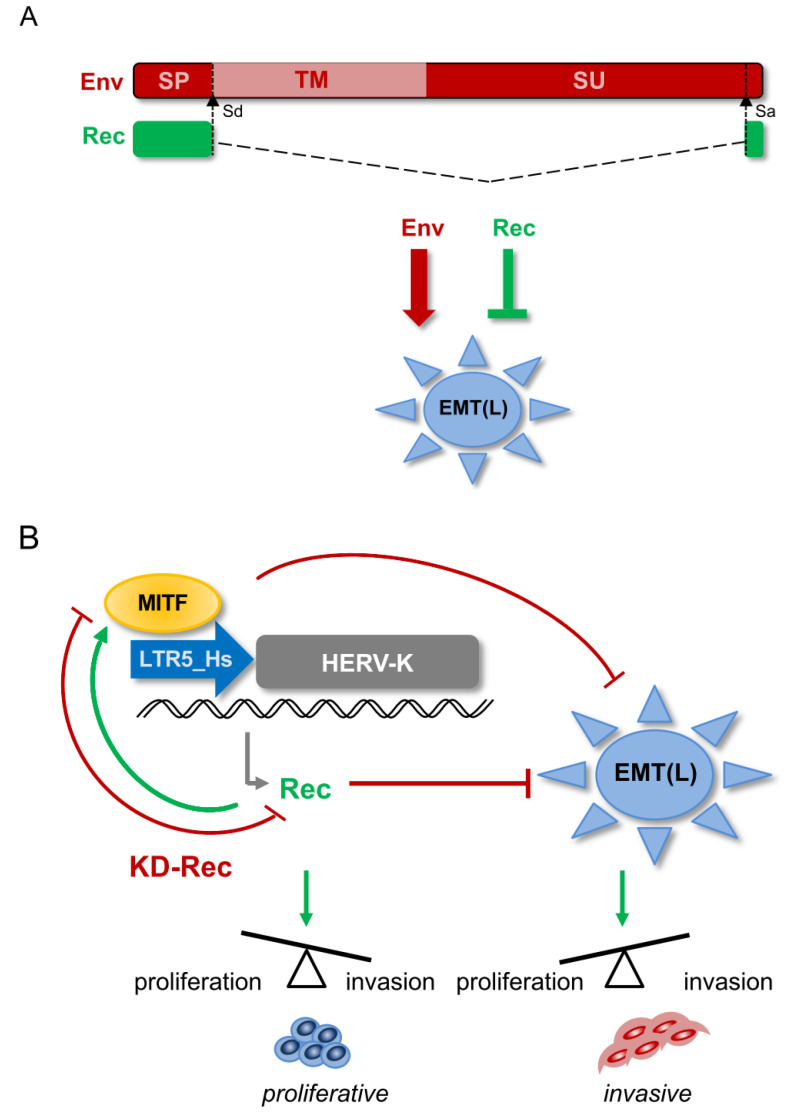
(**A**) Schematic structure of the HERV-K-derived alternative splice products, Env and Rec, and their effect on the endothelial-mesenchymal transition (EMT)-like process of cancer progression EMT(L). The signal peptide (SP); transmembrane subunit (TM); Splice acceptor (Sa); Splice donor (Sd). (**B**) Modeling the interaction loop between MITF and LTR5_Hs and Rec in melanoma. MITF binds to LTR5_Hs and drives Rec expression from certain Type 2 proviruses. Rec marks the proliferative stage melanoma cells and, similarly to MITF, inhibits the endothelial-mesenchymal transition (EMT)-like process EMT(L). Depletion of Rec results in MITF downregulation and the failure to maintain the balance between cell proliferation and invasion. For the limitation of the model see the Discussion.

**Table 1 viruses-12-01303-t001:** The primers/oligos used in this study.

Name	Application	Sequence (5′–3′)	Tm (°C)
HML-2-Env	Real-Time qPCR	F: GCTGCCCTGCCAAACCTGAG R: CCTGAGTGACATCCCGCTTACC	60
HML-2-Gag	Real-Time qPCR	F: AGCAGGTCAGGTGCCTGTAACATT R: TGGTGCCGTAGGATTAAGTCTCCT	60
HML-2-Np9	Real-Time qPCR	F: AGATGTCTGCAGGTGTACCCA R: CTCTTGCTTTTCCCCACATTTC	60
HML-2-Rec	Real-Time qPCR	F: ATCGAGCACCGTTGACTCACAAGA R: GGTACACCTGCAGACACCATTGAT	60
MITF(M)	Real-Time qPCR	F:ATGCTGGAAATGCTAGAATATAATCACT R: GAATGTGTGTTCATGCCTGG	60
SPANXB1/B2	Real-Time qPCR	F: AGGCCAATGAGGCCAACAAGAC R: TCCTCCTGTAGCGAACCACTAG	60
18S RNA	Real-Time qPCR	F: GTAACCCGTTGAACCCCATT R: CCATCCAATCGGTAGTAGCG	60
GAPDH	Real-Time qPCR	F: CAATGACCCCTTCATTGACCTC R: AGCATCGCCCCACTTGATT	60

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
