# Peer review of "Human Endogenous Retrovirus K Rec Forms a Regulatory Loop with MITF that Opposes the Progression of Melanoma to an Invasive Stage"

_viruses, 2020, doi:10.3390/v12111303_

Round 1

Reviewer 1 Report

Viruses

This paper focuses on the role of the HERV-K (HML-12) accessory protein Rec in the progression of melanoma from proliferative to invasive states. They theorize that Rec prevents this transition by acting on EMT mediators such as MITF and limiting expression. This analysis also establishes Rec expression as a marker of this earlier proliferative state.

Major issues

Intro

While correctly pointing out that ERVs were created by ancient viral infection, the intro of this paper fails to fully explain the breadth of proviruses in the human genome. Referring to all LTR5HS proviruses or all HML2 proviruses as just copies of a subgroup fails to accurately capture the differences in sequence between these proviruses. The authors fail to understand or discuss many important and relevant biological facts regarding this group of proviruses, including;

  1. The considerable variation in coding capacity and structure from provirus to provirus, including the fact that of about 900 LTR5 Hs proviruses in the genome, all but about 100 or so are solo LTRs. This becomes important later when understanding the knockdown constructs, since all are directed at 7p22.1, and the importance of letting the reader know which viral proteins are likely to be expressed among the 17 or 18 of interest to the paper.
  2. For a paper that focuses on HML-2 and Rec, the distinction between type 1 and type 2 proviruses is important but not mentioned. Otherwise it is unclear to the reader which LTR5HS proviruses are capable of creating Rec transcripts as opposed to NP9 transcripts (a. protein of unknown function which can’t quite be described as an accessory protein, as it is on line 40).
  3. The often difficult problem of using RNA-seq data to determine which proviruses are expressed due to the similarity of the more recently acquired proviruses to one another (except for deletions).
  4. The fact that the provirus of greatest interest at 7p22.1 (ERVK6—also K108 as the authors should point out but don’t) is actually 2 proviruses in tandem. It should be made clear which of the 2 is referred to when discussing coding capacity.
  5. The role of Rec in HML-2 biology, and how it might apply to direct or indirect effects on the expression of other genes is not presented or discussed.

Results

  1. Although the analysis of chromatin state is an important starting point for this project, showing the wide range of modifications and TF binding sites governing the LTR5HS proviruses across each cell line. It is interesting that there do not seem to be any major changes between cell lines in terms of chromatin marks. The lack of specific provirus IDs is troubling, especially for panel C referring to just LTR5HS. It is unclear if this is a consensus of those LTRs or just a specific provirus. If the latter, at a minimum, they should be separated by major types (1 vs 2, solo LTR vs 2-LTR, etc)
  2. There does seem to be a clear association with the LTR5HS and active chromatin states in numerous cell lines. Based on ChIP results, MITF definitely interacts with and likely plays a role in driving expression of HML-2; however, more definitive data, such as MITF knockdown or direct co transfection, are required to establish it as a critical factor in HML-2 transcription,
  3. It is difficult to understand which proviruses are being expressed in both invasive and proliferative melanoma and what other factors could drive that if we don’t have the sequence.
  4. The Rec knockdown has a clear effect on the transcriptional pattern; yet, again, it focuses on the Rec of 7p22.1 and therefore affects 7 other proviruses as well. The knockdown of Rec does clearly create a change in the cells marked by increased invasiveness and elevation of numerous cellular factors. The GO analysis of the A375 KD cells is not particularly informative, except to show how complex the effects are. This complexity raises a real concern as to which (and whether) the differences observed between the two cell lines are relevant to their different phenotypes, or just idiosyncratic differences from one cell line to the next. Looking at just one cell line does not allow the sort of swweping generalizations made by the authors
  5. Further raising concerns is the oddball effects of Rec KD on the expression of other HML-2 proviruses (Which ones?) in Fig 3F, some of which increase, others decrease. To understand consequences of these changes on protein expression, we have to know which proviruses they are.

Overall, the conclusions drawn regarding the role of Rec in melanoma, and its mecahnism are much too broad and speculative, given the limitations of the study..

Other points:

The manuscript has numerous grammatical and typographic errors that must be corrected in a better-proofread version.

L39-40. Np9 is not an “accessory protein.”

L217 How many of the >600 are solo LTRs?

L234 products of which provirus? It’s incorrect and misleading to make general statements like this as though the phenomenon applied to all HERV-Ks

L 276. Please identify this as aka K108, and nte that it is two tandem oproviruses.

L288-289. Bad phrase.

L300. Which proviruses? How were they identified?

L311-313. Speculative, and uninformative without knowing the proviruses involved.

L389. Do you mean <2e-16?

L394. “multiple” News to me. What other ones besides rec are there?

P8,10. Why the random switch to boldface?

L424-425. Bad phrase.

Figures. Please label the panels and axes more explicitly. It is very confusing to have two different meanings for “CPM.”

Reviewer 2 Report

I have read and reviewed the manuscript titled ‘Human Endogenous Retrovirus K Rec forms a regulatory loop with MITF that opposes the progression of melanoma to an invasive stage’ – here are my comments as a researcher with a decade of experience in the field of epigenetics and transposable elements.

The topic is interesting and the hypothesis attractive – young transposable elements still contain open-reading frames encoding for proteins, and it is intriguing to try to understand how they could impact human health. The authors used a wide variety of public datasets supplemented with their own experiments to try to unveil the contribution of HERV-K (HML2) in modulating the invasiveness phenotype of certain melanoma.

Unfortunately, there are many troubling decisions, inadequate analysis and ill-designed experiments that are glaringly apparent, even at first read, which is very surprising considering the high caliber of both labs involved. While the topic is in itself intriguing, the manuscript in its current form should not be published, and most experiments revised / redone with appropriate controls / reanalyzed. I will not go in the detail (I have many other comments but I don’t think it is useful at this stage) as in my opinion there is a lot of work that needs to be redone before this is ready for publication. I only include some comments aside from the critical concerns to help the authors rethink and rework they experiments, analysis and manuscript. Because many critical flaws were identified, I will not discuss the interpretation, discussion and conclusion that stem from these results, since the identified problems affect almost everything that is presented in the paper.

Critical concerns

  • The use of controls through the manuscript is highly variable, and controls are sometimes grouped together inadequately. For example, for the RNA-seq of the knockdown experiments, it is mentioned that “As an additional control, and to increase statistical power for DEGs identified in our transcriptomes upon Rec-KD, we also used publicly available A375 RNA-seq datasets (GSE110948).” In reality, the authors bundled an unreplicated non-targeting shRNA control, a unreplicated non-transfected control, and a completely unrelated dataset from a different lab also in simplicate, and used that as a ‘triplicate’ to compare to the 3 shRNA data points for their RNA-seq analysis. On top of that, it is even unclear which sh control was used from a public dataset, as the reference cited (GSM122085) refers to a completely different dataset than what is claimed (“Ureaplasma parvum Clinical Isolate” instead of “sh-NS treated A375”). I looked in the list of datasets the authors provide (including the above-cited GSE110948, which does not contain GSM122085), and I could not find where the A375 sh-NS treated control comes from. This control might have used a completely different kit to prepare the RNA-seq library in the best case – did it use the same vector, the same promoter? – but just to be clear, this is a highly inadequate strategy and should be abandoned – sequence additional controls instead of adding random experiments from the Internet, I am certain that the funding from an Advanced ERC grants allows for that. On top of all of this, the use of which control is displayed in figures varies between the bundled controls, shifting between using exclusively the untransfected control, or sometimes an unspecified ‘control’ between panels (Figure 4 panel E for example – why is only the UT control shown?), or the inadequate bundled “control”. Using just an untransfected control is also highly inadequate, especially as a control to transposase-treated, shRNA-producing, FACS-sorted cells. It looks like to me that the experiment was not properly done (i.e control run in triplicate, to match the triplicate bundle of different shRNAs) and authors scrambled to ‘fix’ it, fetching a similar experiment from a public dataset. This is extremely dangerous and makes me very worried that similar ‘creative’ decisions were taken in other steps of the analysis. They should instead show the UT and the scramble controls separately, and avoid doing statistics if their experimental design doesn’t allow for it (and use PCR validation), or redo the experiment to have proper statistical analysis. Also, the supplementary file that contains the RNA-seq data does not contain all the information for all the controls used – it is unclear what the KD-SCR RPKM column represents – the bundled controls or the sh control for the public dataset? Weirdly the UT control is missing
  • A lot of the data originates from the analysis of various NGS datasets. However, the mappability of LTR5_Hs elements is a critical parameter and is never discussed– while it is mentioned that ‘unique reads are used’, this is not sufficient. As I am sure the authors are aware since they are experts on transposable elements, LTR5_Hs being so young it is very hard to map uniquely short reads to individual elements – maybe the borders that are adjacent to other genomic sequences are unique, but internally it is most certainly highly homogeneous. This can lead to a ‘bordering’ effect when using unique mapping which can falsely be detected as ‘peaks’, as reads map to the border, but are discarded from the internal region, leaving a void that combined with the sharp signal edge can be misinterpreted by some peak detection algorithms. Also, the ENCODE datasets used is using 32 bp reads, which is most probably extremely poorly mappable to individual LTR5_Hs. It is not clear if every other datasets analyzed have been truncated down to 32 bp for an homogenous analysis with the ENCODE dataset, or if a variety of read lengths is used between all different analysis. It is unclear if the HMM chromatin model used in figure 1 was built using unique or multi-mapping reads. The authors should provide a mappability analysis of LTR5_Hs, and discuss if it impacts their analysis in any way. This should be supplemented by results of the same analysis unique multi-mapping reads (maybe redistributing of reads according to algorithms such as Expectation Maximization), or a consensus based approach.
  • The authors also include the analysis of a single cell dataset of melanoma. However, they make some critical mistakes in the analysis. The CD248 cluster on the UMAP (cluster 9 - figure 6) is clearly colored red (‘non-malignant’) from the initial annotation of the paper they took it from. Weirdly, it is in every subsequent panels of figure 7 labeled ‘malignant’ – also in the text it is mentioned that there is 5 ‘malignant’ clusters, while in fact there are 4. I think they thought all CD248 cells were the cancer, but the ‘CD248’ labeled cells in cluster 9 are most probably a normal cell type from which the tumor was derived. See “MacFadyen JR, Haworth O, Roberston D, et al. Endosialin (TEM1, CD248) is a marker of stromal fibroblasts and is not selectively expressed on tumour endothelium. FEBS Lett. 2005;579(12):2569‐2575. doi:10.1016/j.febslet.2005.03.071” among others. Weirdly, they don’t show the overlay of CD248 gene signal with other clusters as they do with other markers in the bottom panel, which could have helped. Therefore, most analysis of figure 6 and 7 is completely incorrect. Notwithstanding this, comparing a bundle of various immune cells (lymphocytes and monocytes) to a stromal cancer because they are also ‘non-malignant’ is a very misguided decision in the first place and flaws the analysis in a second very significant way. The heatmap of ‘differentially expressed genes’ between malignant and non-malignant clearly shows this, as the mislabeled ‘malignant’ CD248 dataset clearly clusters with the ‘malignant’ clusters – it is only because they share the same cell type. Redo the analysis with cluster 9 vs cluster 10 and above.

Major comments

  • The claimed differences in chromatin status from the ChromHMM on Ltr5_Hs are not convincing at all – please point out where you identify a significant difference in these patterns, as from my point of view, they are all containing the same mix of chromatin states with slight differences that very well could be random. The use of stacked bars is also potentially misleading, only allowing an easy visual comparison with the top or bottom categories – please use another visualization supplementing this one, for example adding a normal bar chart on the side showcasing the claimed differences in specific categories. Also, the authors state “When compared to differentiated somatic cells, these LTR5_Hs genomic loci showed features of active (‘open’) chromatin state in human embryonic stem cells (H1_ESCs) and in cancer cell lines (Figure 1A).”. Which ones are in the author’s opinion ‘differentiated somatic cells’, and what is the different with ‘cancer cell lines’? Many cell types on the right hand side of the panel, which I assume is what the authors refer to as ‘differentiated somatic cells’, are transformed cell lines – they just transformed naturally, but are cell lines just the same.
  • There should be a clear figure showing where the various shRNA used target in the LTR5_Hs consensus, and if they might also knock down other viral transcripts. Retroviruses are spliced, yielding a complex mix of transcripts that can encode for various proteins – are you sure you are knocking down Rec exclusively in all cases? Are your shRNAs targeting most / all HERVKs equally, or only a subset? Is it the same subset for the three? Any potential off targets (based on sequence similarity) in other ERVs?
  • Figure 1, panel D, it is unclear to me where the annotation for ‘proliferative’ and ‘invasive’ come from. While a paper correlating proliferation and invasiveness states with MITF-F levels is cited, I am not sure if the primary cells analyzed here are the same, or if any evaluation was made of their status. It would be best to only refer them as ‘MITF high’ and ‘MITF low’, while stating in the text that this can be correlated with proliferation status, instead of labeling them ‘invasive’ and ‘proliferating’ exclusively in the subsequent figures. What about the cells with mid-level MITF expression? Why are they not analyzed as well in subsequent figures, and how would they be characterized?
  • Figure 2, panel E should be either percentage-based or preferably enrichment over random expectation based – peak counts as is used can be heavily skewed by different genomic abundance.
  • So many typos in major section of the manuscript that I should be surprised anyone proofread it. Examples include, but are not limited to: line 17 ‘patent’ instead of ‘patient’, line 109 ‘ells’ instead of ‘cells’, line 126 ‘Aligent’ and ‘Aglient’ instead of ‘Agilent’, line 137 ‘Bowite’ instead of ‘Bowtie’, line 145 ‘scambled’ instead of ‘scrambled’ – there is surely more, I stopped looking after that. This does not increase my confidence levels in the carefulness that went in any aspect of the paper, which was already severely impacted by other observations described above.
  • Figure 5 – the exclusive use of an untransfected control in some panels is highly inadequate and flaws the analysis. The author cells have been exposed to a transposase, produce high amounts of shRNAs and GFP, have been sorted (which is a stress that maybe a sub-population of cells survive, which might be the cause of the observed phenotype when compared to wt untransduced unsorted cells). An appropriate sh-scramble or targeting something irrelevant (ie GFP) must be used – having the untransfected control on the side is good, and hopefully will reveal that there’s no impact at all of the vector, transfection and sorting on the analyzed parameters. If the control is unavailable, experiments must be redone de novo.
  • Regarding originality of the research and if appropriate references are cited, please acknowledge previous work that has been done in this exact field (MITF and LTR5_Hs), there’s a few references missing such as Manghera M, Douville RN. Endogenous retrovirus-K promoter: a landing strip for inflammatory transcription factors?. Retrovirology. 2013;10:16. Published 2013 Feb 9. doi:10.1186/1742-4690-10-16

Minor comments

  • Some passages in different font sizes, indicating recent editing by various authors without consolidation prior to submission – please correct
  • Try to keep the order of panels to either left to right and top to bottom – the ordering of panels in figure 2 and 4 is hard to follow
  • Affiliation of C.F. is missing in the header
  • Material and Methods – ‘Construction of the KD constructs’ should contain all the necessary information – some of it is missing (sequence of the scramble short-hairpin used as a control), some of it is disseminated in other sections (vector used). Similarly, the vector that is used (pT2-MP71) has no cited source. Vector used – suitability of MP71 (which I assume is the promoter in absence of any information).

Round 2

Reviewer 1 Report

I was just finishing my review when you cancelled it!

Here it is. My recommendation is to accept after modification.

Reviewer 2 Report

I have read the comments of the authors and reviewed the revised manuscript. I must say I am very pleased with their positive attitude - most points raised by me and the other reviewer were adequately addressed and the manuscript is much improved compared to the original version.

I have a few remaining questions and comments, as well as minor points to address.

Major:

Figure 7 - The comparison being made is in my view still problematic. If you look at the heatmap shown in in Figure 7B, it is clear that we are not looking at genes 'dysregulated in malignant cells'. Categories of 'endothelial' and 'CD248' show the very same transcription profile than genes in the 'malignant category'. What is being done here is the comparison of mostly endothelial cells (above categories + malignant cells) to cell types of the immune system (CD8, CD4, etc). The genes you identify are therefore endothelial specific, not cancer specific. This needs to be corrected / rethought.

Figure 8 (model) First order model with Rec / Gag might be too simple - the authors already make appropriate comments on this in the discussion section, but could soften the language a bit more on their conclusion - what they are really doing is downregulating Rec-containing transcripts, which has a direct impact on the proportion of Gag and Env containing transcripts. Also, althought the authors present some preliminary evidence in this manuscript, it is still unclear if the impact of Rec on cancer proliferation status is correlative or (directly or indirectly) causal.

Minor edits:

line 429 please remove the placeholder "expectation [p-value, test]"

line 484 "LTR5_Hss" instead of "LTR5_Hs"

Figure 1 C - swap ctrl and kd in the figure so they are consistently left or right (choose one) between the figures

figure 3A - It would be nice to see the left boundary of the LTR5_Hs shown to better understand the figure / conclude about the conservation status

figure 4A - The annotations are unclear (which dot is "cell migration" for example?), and there is no X-axis label
